# A theorem on extensive ground state entropy, spin liquidity and some related models

**Sumiran Pujari[1*,2†]**

**1** Department of Physics, Indian Institute of Technology Bombay, Mumbai, MH 400076, India
**2** Max Planck Institute for the Physics of Complex Systems, 01187 Dresden, Germany

* sumiran.pujari@iitb.ac.in , † spujari@pks.mpg.de

## Abstract

An exact mechanism is written down to guarantee extensive residual ground state entropy and spin liquidity in spin-$\frac{1}{2}$ models with bond-dependent couplings. It is based on the presence of extensively large and mutually "«anticommuting»" sets of *local* conserved quantities with a gauge-like character. The general theorem is first pedagogically illustrated through a variant of the familiar one-dimensional quantum Ising model featuring such an «anticommuting» structure that leads to classical spin liquidity co-existing with quantum Ising order. The majority of the paper is then devoted to applications in higher dimensions with more general «anticommuting» structures which voids spin ordering. Proofs of the resultant quantum spin liquidity are given through an analysis of static and dynamic $n$-point spin correlators relying solely on the «anticommuting» algebraic structure of the constructed models. It is not evident if they admit exact solutions using known techniques. The precise nature of these quantum spin liquids is thus an open question including the existence of a quasiparticle description for these models. Discussion is made on this front to compare and contrast with other known cases of quantum spin liquids.

# 1   Introduction

Exact statements are of immense value in quantum many-body physics. They include exactly solvable models of course, but also go beyond them. Well-known examples of the second kind are the Peierls argument for classical Ising models (1; 2) and Elitzur's theorem in the context of lattice gauge theories that forbids local orders (3) with implications for the spontaneous symmetry breaking in superconductors (4). Other semi-rigorous to rigorous examples are the Ginzburg criterion on the validity of mean-field theories (5), and Harris and Imry-Ma criteria on the effect of disorder on clean systems (6; 7; 8; 9).

   In this work, we will make an exact statement of this second kind and illustrate it through various models including solvable ones. The statement concerns a theoretical mechanism that forces an extensive residual ground state entropy on a system along with quantum spin liquidity as the *provable* physical consequences as we shall see. Systems with extensive ground state entropy are often interesting with extremely correlated physics down to the lowest temperatures. Well-known examples are classical spin ices (10) and the SYK model (11). This may seem pathological and in violation of the third law of thermodynamics (12) to a novice in the field of strongly correlated matter. This is rather understood to be the physical state of affairs for generic temperature scales (13) similar to classical spin ices for example. In a "realistic" situation, other (even smaller) couplings will then select the "true" ground state to accord with the third law of thermodynamics often at inaccessible temperatures from a practical point of view. A well-known example of such an inaccessible physics is the prediction of a crystalline state by Wigner (14) in a jellium model of interacting electrons, though there have been other physical situations where this prediction has been realized (15). Here we are not going to concern ourselves with this issue, and are broadly going to focus on the regime of extensive ground state entropy.

## 1.1   Illustrative one dimensional models

With the above motivation, we present the basic ingredients of the theorem in a pedagogical fashion through a variant of the familiar one-dimensional quantum Ising model. Throughout the paper, we will consider ferromagnetic signs for the bond-dependent couplings. Consider the Hamiltonian

$$H = J_x \sum_{\langle i,j \rangle} \sigma_i^x \sigma_j^x + J_z \sum_i \sigma_i^z \sigma_{\partial i}^z \tag{1}$$

where $\partial i$ stands for the auxiliary partner of site $i$ on the spin chain. It is as if we are "applying" the transverse field — of the standard transverse field quantum Ising model (TFQIM) $J \sum_{\langle i,j \rangle} \sigma_i^x \sigma_j^x + h \sum_i \sigma_i^z$ — but now via a transverse Ising coupling of the spins to partner auxiliary spins. Several examples are seen in Fig. 1. We will stick to ferromagnetic couplings throughout in this article without loss of generality.

   Case (a) corresponds to when all sites on the chain obey the unique auxiliary partner condition mentioned in the abstract. Case (b-e) corresponds to when not all sites on the chain

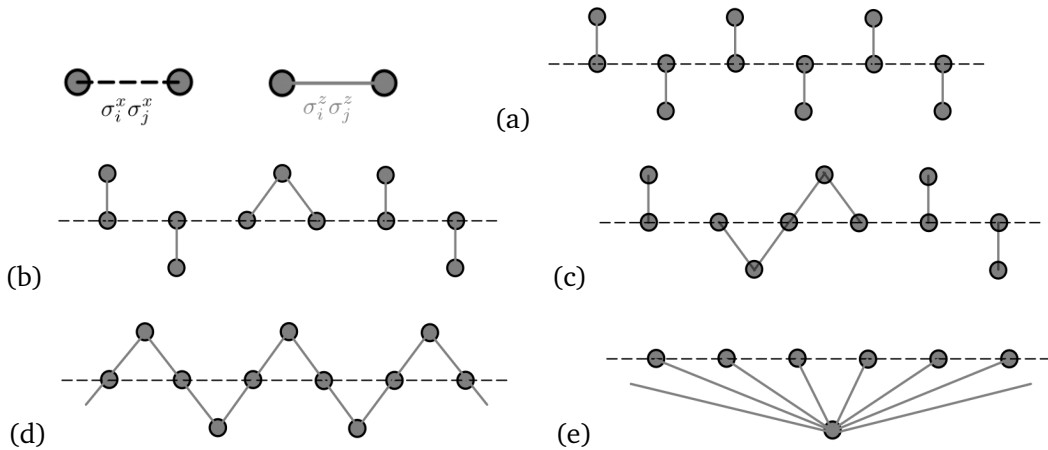

Figure 1: Examples of quantum Ising chains with different configurations for the auxiliary spins.

obey the unique auxiliary partner condition. Case (d-e) corresponds to when all sites on the chain violate the unique auxiliary partner condition.

In all cases, we have the standard global $Z_2$ symmetry of the TFQIM. It may be implemented as a 180° rotation around the $z$-axis, i.e.

$$\mathcal{U}^{Z_2} = \prod_i \otimes \mathcal{R}_i^{\pi,z} \tag{2}$$

with

$$\mathcal{R}_i^{\pi,z} = e^{i\pi\sigma_i^z/2}. \tag{3}$$

Under this

$$\sigma_i^z \to \mathcal{U}^{Z_2}\sigma_i^z\mathcal{U}^{Z_2\dagger} = \mathcal{R}_i^{\pi,z}\sigma_i^z\mathcal{R}_i^{-\pi,z} = \sigma_i^z \tag{4}$$

$$\sigma_i^x \to -\sigma_i^x \tag{5}$$

$$\text{and } H \to H \tag{6}$$

The consequence of this is the conservation of the parity of total chain magnetization in the $z$-direction $M^z = \sum_i \sigma_i^z$. Clearly, $[H, M^z] \neq 0$ but

$$[H, M^z \bmod 2] = \sum_{\langle i,j \rangle}[\sigma_i^x\sigma_j^x, M^z \bmod 2] = 0 \tag{7}$$

This conservation is also equivalent to fermion parity conservation after the Jordan-Wigner transformation (16) which maps the Ising term $\sigma_i^x\sigma_j^x$ to a sum of hopping and superconducting terms in the fermion language. Also in all cases, $\sigma_{\partial i}^z$ is conserved for all $i$, i.e.

$$[H, \sigma_{\partial i}^z] = 0 \tag{8}$$

as can be verified easily. Thus this degree of freedom becomes effectively classical. This in fact facilitates the computation of the exact eigenspectrum via the Jordan-Wigner transformation (17; 18).

Let us start with case (a) in Fig. 1 which satisfies the unique auxiliary partner condition for all sites of the chain. For this case, we have the following:

- The degeneracy of the spectrum is $2^{N_{\partial i}}$ where $N_{\partial i}$ is the number of the auxiliary spins.

- Additionally, the eigenspectrum remains the same as that of the quantum Ising model with $\frac{h}{J} = \frac{J_z}{J_x}$.

We can interpret the above as:

- The auxiliary spins remain paramagnetic down to zero temperature co-existing with Ising order/disorder on the chain (17).

- The presence of an extensive residual entropy or finite residual entropy density.

- One may call this ground state as a co-existence state of Ising order/disorder with a "classical" spin liquid. We will prove this liquidity aspect in Sec. 2.2.

To prove the above we use the following lemma: Let there be two conserved quantities $A$ and $B$, i.e. $[H, A] = [H, B] = 0$, that are mutually anticommuting $\{A, B\} = 0$. For an eigenstate in the $A$-basis, i.e. $H|\psi\rangle = E|\psi\rangle$ and $A|\psi\rangle = a|\psi\rangle$, there exists another state $|B\psi\rangle \equiv B|\psi\rangle$ which is also an eigenstate with $H|B\psi\rangle = E|B\psi\rangle$ and $A|B\psi\rangle = -a|B\psi\rangle$. If $A$ has no zero eigenvalues, then $|B\psi\rangle$ is distinct than $|\psi\rangle$.

The proof goes as follows: There are additional conserved quantities which are absent in the TFQIM. These are $\sigma_i^x \sigma_{\partial i}^x$, i.e.

$$[\sigma_i^x \sigma_{\partial i}^x, H] = 0 \tag{9}$$

for all $i$ as can be verified easily. Furthermore these conserved quantities anticommute with $\sigma_i^z$, i.e.

$$\{\sigma_i^x \sigma_{\partial i}^x, \sigma_{\partial i}^z\} = 0 \tag{10}$$

for all $i$ as can be verified easily.

As nomenclature, we will call sets of local conserved quantities with support over $O(1)$ sites which anticommute when they have sites in common (and commute when no sites in common) as "«anticommuting»" sets of local conserved quantitites. We will also sometimes refer to them as an «anticommuting» structure, an «anticommuting» mechanism, or an «anticommuting» algebra of local conserved quantities. The above conserved sets $\{\sigma_i^x \sigma_{\partial i}^x\}$ and $\{\sigma_{\partial i}^z\}$ form the first example of this structure in this paper.

Also both sets of conserved quantities square to non-zero values and thus have no zero eigenvalues

$$\left(\sigma_i^z\right)^2 = 1 \tag{11}$$

$$\left(\sigma_i^x \sigma_{\partial i}^x\right)^2 = 1 \tag{12}$$

Thus by the application of the lemma above, for each eigenstate $|\psi\rangle$ of $H$, one arrives at $N_{\partial i}$ degenerate eigenstates as $\left(\sigma_i^x \sigma_{\partial i}^x\right)|\psi\rangle$. In fact there are many more degenerate eigenstates arrived at by the operation of the product of $\left(\sigma_i^x \sigma_{\partial i}^x\right)$ over any subset of the auxiliary partner sites. One can convince oneself that the total degeneracy is thus $2^{N_{\partial i}}$.

The eigenspectrum is same as that of TFQIM with $\frac{h}{J} = \frac{J_z}{J_x}$ can be intuitively seen by choosing that sector of the Hamiltonian which corresponds to all the conserved $\sigma_{\partial i}^z$ being all up or all down, i.e. $\prod_{\partial i} \otimes| \uparrow_{\partial i}^z\rangle$ or $\prod_{\partial i} \otimes| \downarrow_{\partial i}^z\rangle$. We formalize this as follows: Let us look at a particular sector or block of the Hamiltonian organized in the basis of conserved quantities $\{\sigma_i^x \sigma_{\partial i}^x\}$, say corresponding to $\langle\sigma_i^x \sigma_{\partial i}^x\rangle = 1$ for all $i$. For each given $\langle\sigma_i^x \sigma_{\partial i}^x\rangle$, there are two compatible states on the bond $(i, \partial i)$. For $\langle\sigma_i^x \sigma_{\partial i}^x\rangle = 1$, we have the states $|\pm_i^x \pm_{\partial i}^x\rangle$ on $(i, \partial i)$ bond. For $\langle\sigma_i^x \sigma_{\partial i}^x\rangle = -1$, we have the states $|\pm_i^x \mp_{\partial i}^x\rangle$. $\sigma_i^z \sigma_{\partial i}^z$ flips between the two compatible states on $(i, \partial i)$ bond. ($\sigma_i^z \sigma_{\partial i}^z|\pm_i^x \pm_{\partial i}^x\rangle = |\mp_i^x \mp_{\partial i}^x\rangle$ for $\langle\sigma_i^x \sigma_{\partial i}^x\rangle = 1$, and $\sigma_i^z \sigma_{\partial i}^z|\pm_i^x \mp_{\partial i}^x\rangle = |\mp_i^x \pm_{\partial i}^x\rangle$ for $\langle\sigma_i^x \sigma_{\partial i}^x\rangle = -1$ respectively.) Thus $\sigma_i^z \sigma_{\partial i}^z$ terms leads to off-diagonal matrix elements for this form of Hamiltonian blocks. If the Hamiltonian blocks were

to be organized using the other conserved set $\{\sigma^z_{\partial i}\}$, then $\sigma^z_i \sigma^z_{\partial i}$ would be a diagonal operator. In our organization of the Hamiltonian blocks using the conserved set $\{\sigma^x_i \sigma^x_{\partial i}\}$, the operator $\sigma^x_i \sigma^x_j$ on the nearest neighbour bonds $(i,j)$ along the chain now measures the parity of the $\sigma^x$-state on these bonds by definition and is thus a diagonal operator. $(\sigma^x_i \sigma^x_j |\pm^x_i \pm^x_j\rangle = +|\pm^x_i \pm^x_j\rangle$ and $\sigma^x_i \sigma^x_j |\pm^x_i \mp^x_j\rangle = -|\pm^x_i \mp^x_j\rangle$ respectively.) Thus Eq. 1 reduces to an effective (dual) TFQIM once the value of $\langle \sigma^x_i \sigma^x_{\partial i}\rangle$ is chosen on all $(i, \partial i)$ bonds. We may write it as follows

$$H = J^{\text{eff}} \sum_{\langle i,j \rangle} \tau^z_i \tau^z_j + h^{\text{eff}} \sum_i \tau^x_i \tag{13}$$

where the $\tau$ operators operate on the two states consistent with $\langle \sigma^x_i \sigma^x_{\partial i}\rangle$, and $J^{\text{eff}} = J_x$, $h^{\text{eff}} = J_z$. As an aside, the parenthetical "dual" refers to the interchanging of the diagonal and off-diagonal operations when organizing the Hamiltonian blocks using the conserved set $\{\sigma^z_{\partial i}\}$ versus the conserved set $\{\sigma^x_i \sigma^x_{\partial i}\}$, while choosing $z$-axis to be the spin-$\frac{1}{2}$ quantization axis as is commonly done.

Now let us consider the case (b) in Fig. 1. Here again we have the conservation of $\sigma^x_i \sigma^x_{\partial i}$ for all $i$ with unique partners. For the two sites which share a partner, the conserved quantity is now $\sigma^x_i \sigma^x_{\partial(i,i+1)} \sigma^x_{i+1}$. This also anticommutes with $\sigma^z_{\partial(i,i+1)}$. Thus we can make similar arguments as above. In case (c), the conserved quantity is $\sigma^x_i \sigma^x_{\partial(i,i+1)} \sigma^x_{i+1} \sigma^x_{\partial(i+1,i+2)} \sigma^x_{i+2}$ with similar physics since in all the above cases (a-c) there are an extensive number of additional conserved quantities. In case (d), the unique partner condition is lost for the full spin chain. Thus in this case we do not have an extensive number of additional conserved quantities. There is only one such quantity, i.e. $\prod_{i,\partial(i,i+1)} \otimes \sigma^x_i \otimes \sigma^x_{\partial(i,i+1)}$. This will lead to a degeneracy of 2 of the spectrum. The configuration of the auxiliary spins which corresponds to the ground state sector also needs determination (17). Case (e) is another such example to show why the physics present in cases (a-c) is absent in the TFQIM. The operator in this case is $\sigma^x_{\partial} \otimes \prod_i \otimes \sigma^x_i$ which is reminiscent of the string operator $\prod_i \otimes \sigma^z_i$ that measures the conserved parity of the magnetization in TFQIM.

Before proceeding further, we note that the above result in the context of quantum Ising models is essentially a restatement of the result obtained in Ref. (18) where the spectral equivalence was rather shown in a converse fashion – modulo minor details – using the set of conserved quantities $\{\sigma^z_{\partial i}\}$ for the Hamiltonian block organization. The context of Ref. (18) was that of quantum compass models with the motivation coming from the physics of orbital degrees of freedom in transition metal systems. The more general structure of "bond algebras" underlying this work is further elaborated in Refs. (19), including resultant dualities in Refs. (20; 21) an example of which we saw as Eq. 13 in the analysis of Eq. 1 above. Furthermore the lemma above and its variants have been used previously in other contexts (22), including the context of ground state degeneracy and topological orders. See in particular Ref. (23) which gives a good overview of the existing results in the literature. The attention in these works has been on sub-extensive degeneracies and, generally speaking, topological (or non-topological) quantum orders that have ground state manifolds with zero ground state entropy density which is not the focus of this work. The model Hamiltonians that form the main subject of this paper (Sec. 2) are also Nussinov-Ortiz bond algebras (18; 19). However, the ingredient of the «anticommuting» algebraic structure of extensively many local conserved quantities leads to the new physics reported in this paper, i.e. finite ground state entropy density and quantum spin liquidity. We will compare and contrast in detail this physics with known quantum spin liquids in Sec. 2.3.

### 1.2 Further discussion

Let us consider case (a) for this discussion. It is natural to block diagonalize the Hamiltonian $H$ in terms of the conserved spin configurations of the auxiliary partner spins $\prod_{\partial i} \otimes |\sigma^z_{\partial i}\rangle$. However, the conservation of $\sigma^x_i \sigma^x_{\partial i}$ begs the following question: How to understand the physics if we were to organize the Hamiltonian blocks in terms of the conserved $\sigma^x_i \sigma^x_{\partial i}$ for all $i$ ? Firstly, fixing the configuration of the auxiliary spins as $\prod_{\partial i} \otimes |\sigma^z_{\partial i}\rangle$ implies no fluctuation in them. But fixing the eigenvalues (of $\pm 1$) of the conserved $\sigma^x_i \sigma^x_{\partial i}$ for all $i$ does not imply any such thing. In this way of block diagonalization, both the spins of the spin chain and the auxiliary spins keep fluctuating. This suggests that the (local) conservation of $\sigma^x_i \sigma^x_{\partial i}$ has a gauge-like character. From this point of view, for a given eigenstate $|\psi\rangle$, we can obtain degenerate eigenstates as $\sigma^z_{\partial i} |\psi\rangle$ or $\prod_{\{\partial j\} \subseteq \{\partial i\}} \sigma^z_{\partial j} |\psi\rangle$ for any subset of auxiliary spins. This again gives a degeneracy of $2^{N_{\partial i}}$ as expected. Due to this extensive degeneracy, the gauge charges or eigenvalues of the conserved $\sigma^x_i \sigma^x_{\partial i}$ can also keep fluctuating. This is because we can linearly combine the eigenstates from different gauge charge sectors to obtain a new eigenstate. Under time evolution, this linear combination will stay put, i.e. both the gauge charges and the auxiliary spin states keep fluctuating for all times.

Another perspective is to look at the same physics after the Jordan-Wigner transformation. Then we arrive at

$$H = J_x \sum_{\langle i,j \rangle} \left( c^\dagger_i c_j + c^\dagger_i c^\dagger_j + \text{h.c.} \right) + J_z \sum_i (2n_i - 1)(2n_{\partial i} - 1) \tag{14}$$

The global fermion parity is again conserved due to global $Z_2$ symmetry, but we also have local $Z_2$ symmetries in terms of local 180° rotations around the $x$-axis for site $i$ and $\partial i$ which keep the Hamiltonian unchanged. This implies the conservation of $\sigma^x_i \sigma^x_{\partial i}$ on $(i, \partial i)$ bonds. Upon Jordan-Wigner transformation, we get

$$[H, (-1)^{\text{Jordan-Wigner phases}} \left( c^\dagger_i c_{\partial i} + c^\dagger_i c^\dagger_{\partial i} + \text{h.c.} \right)] = 0. \tag{15}$$

However, there are no kinetic hopping or superconducting terms $\propto \left( c^\dagger_i c_{\partial i} + c^\dagger_i c^\dagger_{\partial i} + \text{h.c.} \right)$ corresponding to the local $Z_2$ charges on $(i, \partial i)$ bonds. Thus all gauge sectors are degenerate. Also the mutual anticommutation of

$$\{ c^\dagger_i c_{\partial i} + c^\dagger_i c^\dagger_{\partial i} + \text{h.c.}, n_{\partial i} \} = 0 \tag{16}$$

implies that local $Z_2$ charges can fluctuate along with $n_{\partial i}$.

### 1.3 Some Extensions

Let us continue with case (a). Since $\sigma^x_i \sigma^x_{\partial i}$ is conserved, the following Hamiltonian

$$H = J_x \sum_{\langle i,j \rangle} \sigma^x_i \sigma^x_j + J_z \sum_i \sigma^z_i \sigma^z_{\partial i} + J'_x \sum_i \sigma^x_i \sigma^x_{\partial i} \tag{17}$$

also is solvable. However $\sigma^z_{\partial i}$ is not conserved anymore. Thus the extensive degeneracy will be lost. The spectrum now will depend on the conserved value of $\sigma^x_i \sigma^x_{\partial i}$ on all $(i, \partial i)$ bonds. E.g. the ground state will correspond to $\langle \sigma^x_i \sigma^x_{\partial i} \rangle = 1$ for ferromagnetic $J'_x < 0$. Following same arguments on spectral equivalence of Eq. 1 and the TFQIM as before in Sec. 1.1 (see the discussion around Eq. 13), the above Hamiltonian also reduces to an effective (dual) TFQIM once the value of $\langle \sigma^x_i \sigma^x_{\partial i} \rangle$ is chosen on all $(i, \partial i)$ bonds. We may write it as follows

$$H = J^{\text{eff}} \sum_{\langle i,j \rangle} \tau^z_i \tau^z_j + h^{\text{eff}} \sum_i \tau^x_i + J'_x \sum_i \langle \sigma^x_i \sigma^x_{\partial i} \rangle \tag{18}$$

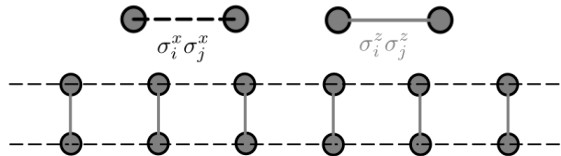

Figure 2: Ladder geometry with bond-dependent couplings as discussed in the text.

where the $\tau$ operators operate on the two states consistent with $\langle \sigma_i^x \sigma_{\partial i}^x \rangle$ as befpre, and $J^{\text{eff}} \propto J_x$, $h^{\text{eff}} \propto J_z$. One will again obtain the TFQIM spectrum in any conserved sector. The loss of extensive degeneracy corresponding to $J_x' = 0$ is seen through the third term above $J_x' \sum_i \langle \sigma_i^x \sigma_{\partial i}^x \rangle$. One sees that there are still degenerate excited sectors given by different configurations of $\langle \sigma_i^x \sigma_{\partial i}^x \rangle$ which keep the sum $\sum_i \langle \sigma_i^x \sigma_{\partial i}^x \rangle$ fixed. The degeneracies are basically $N_i$ choose $N_{\langle \sigma_i^x \sigma_{\partial i}^x \rangle = 1}$. These degeneracies will likely be further broken down in presence of additional solvability breaking terms. By a similar token, for the following Hamiltonian

$$H = J_x \sum_{\langle i,j \rangle} \sigma_i^x \sigma_j^x + J_z' \sum_{\langle i,j \rangle} \sigma_i^z \sigma_j^z + J_z \sum_i \sigma_i^z \sigma_{\partial i}^z \tag{19}$$

$\sigma_i^x \sigma_{\partial i}^x$ is not conserved anymore, but $\sigma_{\partial i}^z$ stays conserved. Thus the extensive degeneracy will again be lost. However, solving for the spectrum using the Jordan-Wigner transformation is not that straightforward.

Also may be noted that the following related ladder Hamiltonian

$$H = J_x \sum_{\langle i,j \rangle} \sigma_i^x \sigma_j^x + J_{\partial x} \sum_{\langle i,j \rangle} \sigma_{\partial i}^x \sigma_{\partial j}^x + J_z \sum_i \sigma_i^z \sigma_{\partial i}^z + J_x' \sum_i \sigma_i^x \sigma_{\partial i}^x \tag{20}$$

as shown in Fig. 2 is effectively equivalent to

$$H = \sum_{\langle i,j \rangle} J_{ij}^{\text{eff}} \tau_i^z \tau_j^z + h^{\text{eff}} \sum_i \tau_i^x + J_x' \sum_i \langle \sigma_i^x \sigma_{\partial i}^x \rangle \tag{21}$$

with $h^{\text{eff}} \propto J_z$. The case of $J_{ij}^{\text{eff}}$ requires more attention. For the (ground state) sector corresponding to $\langle \sigma_i^x \sigma_{\partial i}^x \rangle = 1$ on all $(i, \partial i)$ bonds, $J_{ij}^{\text{eff}} \propto (J_x + J_{\partial x})$ independent of the bond location. The same would be true for the sector corresponding to $\langle \sigma_i^x \sigma_{\partial i}^x \rangle = -1$ on all $(i, \partial i)$ bonds. Recall we are considering ferromagnetic couplings in this article throughout. For other sectors where $\langle \sigma_i^x \sigma_{\partial i}^x \rangle$ is not uniformly the same sign, the bond location becomes important. For a bond $(i, j)$ such that $\langle \sigma_i^x \sigma_{\partial i}^x \rangle = \langle \sigma_j^x \sigma_{\partial j}^x \rangle$, $J_{ij}^{\text{eff}} \propto (J_x + J_{\partial x})$. For a bond $(i, j)$ such that $\langle \sigma_i^x \sigma_{\partial i}^x \rangle \neq \langle \sigma_j^x \sigma_{\partial j}^x \rangle$, $J_{ij}^{\text{eff}}$ itself fluctuates between $\propto (J_x - J_{\partial x})$ and $\propto -(J_x - J_{\partial x})$ depending on the state of the spins on the $(i, \partial i)$ and $(j, \partial j)$ bonds. Obtaining the spectrum in these excited sectors is therefore more involved. For $J_x = J_{\partial x}$ (24) which would be the case in presence of mirror symmetry between the two legs of the ladder, there occurs a simplification and $J_{ij}^{\text{eff}} = 0$ on those bonds where $\langle \sigma_i^x \sigma_{\partial i}^x \rangle \neq \langle \sigma_j^x \sigma_{\partial j}^x \rangle$. This leads to disconnected TFQIM segments which can again be solved for the excited eigenspectrum. The resultant physics has been more generally termed as Hilbert space fragmentation (25). The above ladder Hamiltonian can be looked at as a solvable local spinless fermionic model for $J_x' = 0$,

$$H = -t_x \sum_{\langle i,j \rangle} (c_i^\dagger c_j + c_i^\dagger c_j^\dagger + \text{h.c.}) - t_{\partial x} \sum_{\langle i,j \rangle} (c_{\partial i}^\dagger c_{\partial j} + c_{\partial i}^\dagger c_{\partial j}^\dagger + \text{h.c.}) + V \sum_i n_i n_{\partial i} \tag{22}$$

The physical situation is that of two $p$-wave superconducting wires coupled through a short-ranged Couloumb interaction. Analogous physics will carry through in this context. We may

conjecture that the physics extends to situation when the hopping and pairing amplitudes are not exactly equal. A stronger conjecture would be the stability of the ground state in presence of hopping and/or superconducting amplitudes between the two wires. The ground state of the fermionic system will be two locked superconducting ground states independent of $V$. The solvable case of $J'_x \neq 0$ leads to non-local terms in the fermionic situation and may be ignored.

## 2  Two-dimensional constructions

Till now the discussion has been limited to one dimensional systems. Let us now construct two-dimensional spin models with extensive ground state entropy guaranteed through the mechanism underlying the theorem, i.e. existence of extensively large mutually «anticommuting» sets of conserved quantities. To repeat the nomenclature introduced in Sec. 1.1, by «anticommuting» sets, we mean the anticommutation of two conserved operators coming from different sets when they have a common site between them as in the previous Sec. 1.1 and throughout the paper. The condition of a single common site is however not strict, even though respected in all the models discussed in this paper. One can easily construct variants where one may have the «anticommuting» mechanism operational even when local conserved quantities coming from different sets have more than one site in common. Conserved quantities without common sites of course keep commuting in the context of spin models. We continue to take ferromagnetic signs for the bond-dependent couplings.

### 2.1  Generic degeneracy counting

Consider the bond-dependent Hamiltonian

$$H = J_x \sum_{\boxed{x}} \left( \sum_{\langle i,j \rangle \in \boxed{x}} \sigma_i^x \sigma_j^x \right) + J_z \sum_{\boxed{z}} \left( \sum_{\langle i,j \rangle \in \boxed{z}} \sigma_i^z \sigma_j^z \right) \tag{23}$$

sketched in Fig. 3. The system is composed of square plaquettes with either $\sigma_i^x \sigma_j^x$ or $\sigma_i^z \sigma_j^z$ couplings exclusively arranged in a checkerboard pattern. $\boxed{x}$ and $\boxed{z}$ denote the plaquettes with $\sigma_i^x \sigma_j^x$ and $\sigma_i^z \sigma_j^z$ respectively. Let us look at the conserved quantities. They are

1. $\sigma_i^z \sigma_j^z \sigma_k^z \sigma_l^z$ on the $\boxed{x}$ plaquettes.

2. $\sigma_i^x \sigma_j^x \sigma_k^x \sigma_l^x$ on the $\boxed{z}$ plaquettes.

The conserved nature of these quantities may be verified easily. All the above quantities form extensively large sets due to their local nature. The two sets «anticommute» with each other.
    Eigenspectrum solvability of this model is not apparent, but by the application of the lemma, we can conclude that this system will host an extensive ground state entropy. In fact, the full eigenspectrum will be massively degenerate in this sense. The counting can be ascertained by first spanning the system with one of the conserved sets from the above that can serve as the basis for block-diagonalization of the Hamiltonian, and then counting the other set that «anticommutes» with the chosen set. Thus the ground state entropy is $\ln 2$ per unit cell. A related Hamiltonian with the same degeneracy counting could be constructed using the above conserved operators directly,

$$H = J_x \sum_{\boxed{x}} \left( \prod_{i \in \boxed{x}} \otimes \sigma_i^x \right) + J_z \sum_{\boxed{z}} \left( \sum_{i \in \boxed{z}} \otimes \sigma_i^z \right) \tag{24}$$

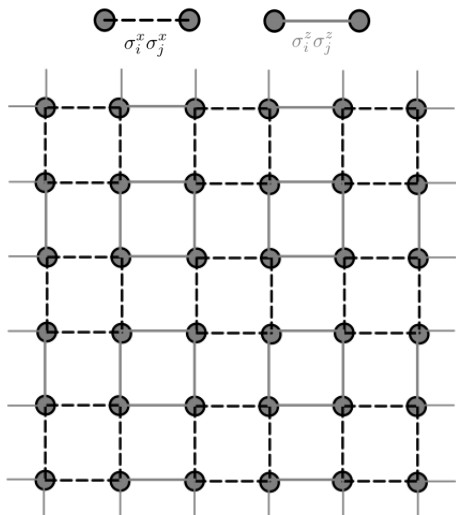

Figure 3: A two-dimensional spin model with extensive ground state entropy.

following the spirit of Kitaev toric code (26) whose Hamiltonian is also the sum of the conserved quantities which however mutually commute. We will stick to models with two-spin terms without any loss of generality.

An alternative Hamiltonian with this anticommuting mechanism in operation is shown in Fig. 4. The system is now composed of crisscrossing Ising chains with couplings in perpendicular directions in spin space. It is a square lattice variant of the Kitaev honeycomb model (27) and in fact belongs to the class of "compass" models (28; 29). Its ground state properties have been discussed in the literature (30; 31; 32). It can be written as

$$H = \sum_{\mathbf{r}} J_x \sigma_{\mathbf{r}}^x \sigma_{\mathbf{r}+\mathbf{e_x}}^x + J_z \sigma_{\mathbf{r}}^z \sigma_{\mathbf{r}+\mathbf{e_z}}^z \tag{25}$$

The degeneracy counting in this model has also been done through the lens of the anticommuting mechanism, however the it only leads to a double degeneracy independent of system size (30). This is because the conserved $Z_2$ parities for each Ising chain are *non-local* string operators as in TFQIM: $\prod_{r_y} \sigma_{(r_x, r_y)}^x$ for a given $r_x$ and $\prod_{r_x} \sigma_{(r_x, r_y)}^z$ for a given $r_y$. Note the number of these non-local conserved quantities is sub-extensive and not extensive in contrast to the other models. Due to the geometry of the string operators – each string operator in a given direction intersects all string operators in the perpendicular direction – the application of the lemma only gives a double degeneracy. An interesting counterpoint in the context of this paper is the following: even though the degeneracy is $O(1)$ by the anticommuting mechanism, it has been stated by Dorier *et al* that, "When $J_x \neq J_z$, we show that, on clusters of dimension $L \times L$, the low-energy spectrum consists of $2^L$ states which collapse onto each other exponentially fast with $L$, a conclusion that remains true arbitrarily close to $J_x = J_z$. At that point, we show that an even larger number of states collapse exponentially fast with $L$ onto the ground state, and we present numerical evidence that this number is precisely $2 \times 2^L$." (31) It is as if the system "would prefer" a (sub-extensively) large degeneracy, however there are no symmetries to guarantee it exactly. In all the other constructions discussed in this paper, we can guarantee rather an extensive degeneracy because of the local nature of the conserved quantities, i.e. they have support on $O(1)$ lattice sites. Furthermore, additional degeneracies at "fine-tuned" coupling values might be present even in these models, but we are not concerning ourselves with such effects in this paper. The generic extensive entropy case afforded by the anticommuting mechanism of this paper can already be used to prove spin liquid nature of these models as will be discussed in the next Sec. 2.2. Another example is a one-dimensional

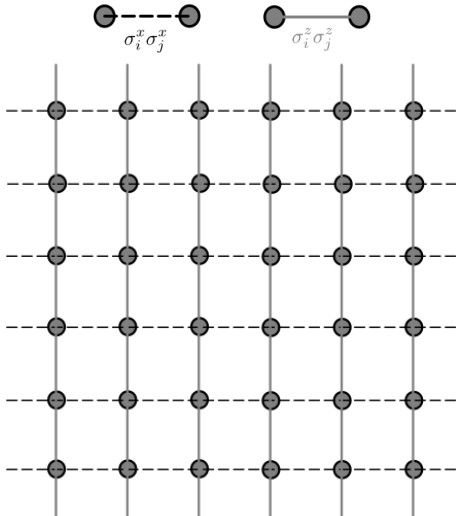

Figure 4: The 90° compass model with only a double degeneracy via the anticommuting mechanism. The low-energy manifold is only sub-extensive in size, which leads to a zero low-energy entropy density in contrast to the other cases studied in this paper.

version of the compass model (33) where, by virtue of the reduced dimensionality, the conserved quantities become local and the theorem then guarantees an extensive degeneracy even for finite chains (34). It can also be reduced to an effective TFQIM and follows the spirit of the one-dimensional cases discussed earlier.

Consider finally the Hamiltonian in Fig. 5 which has a more intricate structure compared to the previous models. This case is of interest because of the technical differences in its ground state entropy counting compared to earlier which might be worth pointing out. The system is now composed of a) square plaquettes with either $\sigma_i^x \sigma_j^x$ or $\sigma_i^z \sigma_j^z$ couplings exclusively denoted as $\boxed{x}$ and $\boxed{z}$ respectively, b) crosses (×) composed of both $\sigma_i^x \sigma_j^x$ and $\sigma_i^z \sigma_j^z$ segments crisscrossing each other, and c) hexagonal plaquettes with alternating $\sigma_i^x \sigma_j^x$ and $\sigma_i^z \sigma_j^z$ cou-

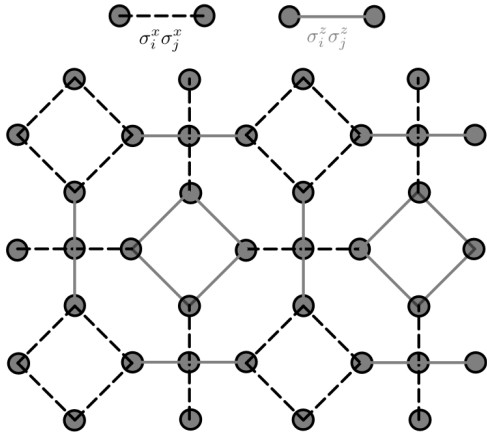

Figure 5: A more intricate two-dimensional spin model with extensive ground state entropy.

292  plings as a result of a) and b). It can formally be written as

$$H = J_x \sum_{\boxed{x}} \sum_{\langle i,j \rangle \in \boxed{x}} \sigma_i^x \sigma_j^x + J_z \sum_{\boxed{z}} \sum_{\langle i,j \rangle \in \boxed{z}} \sigma_i^z \sigma_j^z + J_x' \sum_{\times} \sum_{\langle i,j \rangle_x \in \times} \sigma_i^x \sigma_j^x + J_z' \sum_{\times} \sum_{\langle i,j \rangle_z \in \times} \sigma_i^z \sigma_j^z$$
(26)

293  The conserved quantities are

294  1. $\sigma_i^z \sigma_j^z \sigma_k^z \sigma_l^z$ on the $\boxed{x}$-plaquettes.

295  2. $\sigma_i^x \sigma_j^x \sigma_k^x \sigma_l^x$ on the $\boxed{z}$-plaquettes.

296  3. $\sigma_i^z \sigma_j^z \sigma_k^z$ on the bonds $\langle i,j \rangle_x \in \times$.

297  4. $\sigma_i^x \sigma_j^x \sigma_k^x$ on the bonds $\langle i,j \rangle_z \in \times$.

298  The conserved nature of these quantities may be verified easily. There does not seem to be any
299  obvious conserved quantity associated with the hexagonal plaquettes. All the above quantities
300  form extensively large sets. The first and second sets of conserved quantities commute with
301  each other. The third and fourths sets «anticommute» with each other. Similarly, the first and
302  fourth sets anticommute with each other and the second and third sets «anticommute» with
303  each other. The first and third sets commute with each other, and the second and fourth sets
304  commute with each other.

305  Similar to Eq. 23, eigenspectrum solvability of the model in Eq. 26, Fig. 5 is not apparent
306  and there will be a massive degeneracy of its eigenspectrum. The counting can be ascertained
307  by first spanning the system with mutually conserved sets from the above, and then counting
308  the remaining sets that «anticommute» with the spanning sets. The maximum of all possible
309  ways of doing this will give the entropy due to this mechanism. In this model, if we use
310  the first and third sets as the spanning sets, then the remaining sets contribute an entropy of
311  $3 \ln 2\, k_B$ per unit cell. Similarly, if we use second and fourth sets as the spanning sets, then
312  the remaining sets again contribute an entropy of $3 \ln 2\, k_B$ per unit cell. Instead, if we use the
313  first and second sets as the spanning sets, then the remaining sets contribute an entropy of
314  $4 \ln 2\, k_B$ per unit cell. The ground state entropy is therefore $4 \ln 2\, k_B$ per unit cell through this
315  anticommuting mechanism.

316  There is an unresolved puzzle with respect to the above degeneracy counting even though
317  it is exact and non-perturbative. If we were to think of Eq. 26 through a perturbative lens,
318  then there are two natural ways of going about it. If we take all the terms with $\sigma_i^x \sigma_j^x$ Ising
319  couplings as the dominant terms and the terms with $\sigma_i^z \sigma_j^z$ Ising couplings as the perturbation
320  or vice versa, one arrives at an entropy of $3 \ln 2$ per unit cell. On the other hand, if we
321  take all the terms involving the boxed plaquettes $\boxed{x}$ and $\boxed{z}$ as the dominant terms with
322  the rest being the perturbation, then we arrive at an entropy of $4 \ln 2$ per unit cell. Since 4
323  $\ln 2$ is the non-perturbative count, the additional $\ln 2$ contribution is not accounted for when
324  setting up the perturbation theory in the first manner. It is a puzzle as to how this additional
325  degeneracy would be accounted for at all orders in perturbation theory when doing it in this
326  manner. Note that additional degeneracies can arises at special points as pointed out by Dorier
327  *et al* (31), however here it must happen without any such fine-tuning, i.e. for any value of the
328  perturbation.

329  On a related note, for the model of Eq. 23, in the limit of $J_x = 0$ or $J_z = 0$ the degeneracy
330  corresponding to $\ln 2$ entropy per unit cell is obvious. How this degeneracy survives to all
331  orders in perturbation theory when going away from these limits is another related question.
332  The answer must be of the form that no perturbation at any order can connect two degenerate
333  states to split the degeneracy. For an example of such an effect, see e.g. Ref. (35). This is

Table 1: A generic set $A$ of 4 ground states that form an equivalence class given two sites $i_\times$ and $j_\times$ for the model of Eq. 26 and Fig. 5.

| $\|\psi\rangle$ | $\langle\psi\|\sigma^z_{\partial_1 i_\times} \sigma^z_{i_\times} \sigma^z_{\partial_2 i_\times}\|\psi\rangle$ | $\langle\psi\|\sigma^z_{\partial_1 j_\times} \sigma^z_{j_\times} \sigma^z_{\partial_2 j_\times}\|\psi\rangle$ | $\langle\psi\|\sigma^\mu_{i_\times} \sigma^\nu_{j_\times}\|\psi\rangle$ |
|---|---|---|---|
| $\|\psi^A_{\mathrm{gs}}\rangle$ | +1 | +1 | $\langle\psi^A_{\mathrm{gs}}\|\sigma^\mu_{i_\times} \sigma^\nu_{j_\times}\|\psi^A_{\mathrm{gs}}\rangle$ |
| $\sigma^x_{\partial_3 i_\times} \sigma^x_{i_\times} \sigma^x_{\partial_4 i_\times}\|\psi^A_{\mathrm{gs}}\rangle$ | -1 | +1 | $(2\delta_{\mu x} - 1)\langle\psi^A_{\mathrm{gs}}\|\sigma^\mu_{i_\times} \sigma^\nu_{j_\times}$ |
| $\sigma^x_{\partial_3 j_\times} \sigma^x_{j_\times} \sigma^x_{\partial_4 j_\times}\|\psi^A_{\mathrm{gs}}\rangle$ | +1 | -1 | $(2\delta_{\nu x} - 1)\langle\psi^A_{\mathrm{gs}}\|\sigma^\mu_{i_\times} \sigma^\nu_{j_\times}$ |
| $\sigma^x_{\partial_3 i_\times} \sigma^x_{i_\times} \sigma^x_{\partial_4 i_\times} \sigma^x_{\partial_3 j_\times} \sigma^x_{j_\times} \sigma^x_{\partial_4 j_\times}\|\psi^A_{\mathrm{gs}}\rangle$ | -1 | -1 | $(2\delta_{\mu x} - 1)(2\delta_{\nu x} - 1)\langle\psi^A_{\mathrm{gs}}\|\sigma$ |

a different question from the earlier puzzle. In the earlier puzzle, we are asking how *extra* degeneracies are *generated* at all orders in perturbation theory set up in a particular way. In the above question, we are asking how the *already present* degeneracies are *preserved* to all orders in perturbation theory.

## 2.2 Quantum Spin liquidity

We will now prove ground state spin liquidity in the extensively degenerate models using just the anticommuation structure. For example for the model of Fig. 5 or Eq. 26, there are three kinds of sites: sites at the centre of the crosses ($i_\times$), those on the $\boxed{x}$-plaquettes ($i_x$) and those on the $\boxed{x}$-plaquettes ($i_z$). The proof can be understood by taking one representative example, say the ground state expectation $\langle\sigma^\mu_{i_\times} \sigma^\nu_{j_\times}\rangle$ on two different faraway sites. This ground state expectation value is to be understood as a thermal mixture over the ground state manifold as $T \to 0$, i.e.

$$\langle O\rangle(T \to 0) = \sum_{|\psi\rangle \in \{|\psi_{\mathrm{gs}}\rangle\}} \langle\psi|O|\psi\rangle \tag{27}$$

where $\{|\psi_{\mathrm{gs}}\rangle\}$ is the ground state manifold.

Working in the basis of the first and third commuting sets ("$z$"-basis), we can subdivide the ground state manifold into distinct sets or "equivalence classes" containing 16 ground states each given the two unit cells to which the sites $i_\times$ and $j_\times$ belong. If a generic set $A$ is indexed by a representative ground state $|\psi^A_{\mathrm{gs}}\rangle$, then we can generate the other 15 ground states by the application of $\sigma^x_i \sigma^x_j \sigma^x_k \sigma^x_l$ and the two different $\sigma^x_i \sigma^x_j \sigma^x_k$ belonging to the two unit cells on $|gs_A\rangle$. (16=1+3+3+(3×3).) If we sum $\langle\sigma^\mu_{i_\times} \sigma^\nu_{j_\times}\rangle$ over all these sixteen states, one finds that the sum is zero for all cases of $\mu, \nu$ except for $\langle\sigma^x_{i_\times} \sigma^x_{j_\times}\rangle$. To show that the sum is zero even in this case, one can rework the above starting from "$x$"-basis involving the second and fourth sets. Thus this will be true for the overall ground state manifold sum.

We will present a simpler argument below by only involving the conserved operators that include the sites $i_\times$ and $j_\times$ which would lead to a set of 4 ground states. The division into the set of 16 related ground states organized by unit cells is somewhat more natural. Working in the "$z$"-basis, let the representative state $|\psi^A_{\mathrm{gs}}\rangle$ correspond to the value of +1 for the conserved quantities $\sigma^z_{\partial_1 i_\times} \sigma^z_{i_\times} \sigma^z_{\partial_2 i_\times}$ and $\sigma^z_{\partial_1 j_\times} \sigma^z_{j_\times} \sigma^z_{\partial_2 j_\times}$ connected to the two sites $i_\times$ and $j_\times$. The set $A$ can be indexed by the values of all the other conserved quantities in the "$z$"-basis *given* the previous statement. We arrive at Table 1 after generating the set of 4 states. The table shows that this grouping of the ground state manifold is unique. This is because the three states generated from $|\psi^A_{\mathrm{gs}}\rangle$ can not correspond by construction to a different representative state since the value of $\sigma^z_{\partial_1 i_\times} \sigma^z_{i_\times} \sigma^z_{\partial_2 i_\times}$ and $\sigma^z_{\partial_1 j_\times} \sigma^z_{j_\times} \sigma^z_{\partial_2 j_\times}$ are not +1 for these three states. Clearly the sum over these 4 states is zero whenever $\mu \neq x$ or $\nu \neq x$. For $\langle\sigma^x_{i_\times} \sigma^x_{j_\times}\rangle$, we start in the "$x$"-basis and redo the above as mentioned before. The associated table would be the same as Table 1 with the interchanging of $x$ and $z$ everywhere.

One can similarly argue for the vanishing of "2-point" spin order when $i_x$ or $i_z$ type of sites are involved. Also, one can see from these arguments that the "faraway" requirement of the two sites is not very strict. This is analogous to the Kitaev model (36; 37) however guaranteed through the anticommuting mechanism. In fact, as we see, we did not need any other representation (fermionic or otherwise) to prove this which highlights the reach of the anticommuting structure. Furthermore, one can extend these arguments to "$n$-point" spin orders involving different unit cells. A similar argument goes through for the model in Eq. 23 and Fig. 3 which is composed of only one kind of lattice site.

Multi-spin order parameter correlations such as bond energies, plaquette spin products, etc. can survive the above cancellations. Let us restrict ourselves to the situation of faraway unit cells and the model of Eq. 23 for simplicity. We will state the result without giving the proof. The proof logic follows from the above sort of arguments. For a 2-spin operator $\sigma_i^\mu \sigma_j^\nu$ on a bond $\langle i, j \rangle$, its 2-bond correlators $\langle \left( \sigma_i^\mu \sigma_j^\nu \right) \left( \sigma_k^\gamma \sigma_l^\delta \right) \rangle$ are non-zero only when $\{\mu, \nu, \gamma, \delta\}$ correspond to the spin space indices that *appear* in the Hamiltonian on their corresponding sites $\{i, j, k, l\}$. E.g., for bonds $\langle i, j \rangle$ and $\langle k, l \rangle$ which host $\sigma_i^x \sigma_j^x$ and $\sigma_k^x \sigma_l^x$, the only non-zero correlator is $\langle \left( \sigma_i^x \sigma_j^x \right) \left( \sigma_k^x \sigma_l^x \right) \rangle$ while all other correlators are zero through the above kind of cancellation arguments. Similarly for bonds $\langle i, j \rangle$ and $\langle k, l \rangle$ which host $\sigma_i^x \sigma_j^x$ and $\sigma_k^z \sigma_l^z$, the only non-zero correlator is $\langle \left( \sigma_i^x \sigma_j^x \right) \left( \sigma_k^z \sigma_l^z \right) \rangle$ and so on.

For a 3-spin and higher-spin operators, one has to pay a little more care. The 3-spin case exposes the general structure of these multi-spin correlations. For $\sigma_i^\mu \sigma_j^\nu \sigma_k^\gamma$ on 3 contiguous sites $\langle i, j, k \rangle$, its 2-bond correlators $\langle \left( \sigma_i^\mu \sigma_j^\nu \sigma_k^\gamma \right) \left( \sigma_l^\delta \sigma_m^\alpha \sigma_n^\beta \right) \rangle$ are non-zero only when $\{\mu, \nu, \gamma, \delta, \alpha, \beta\}$ correspond to the spin space indices that "appear" in the Hamiltonian on the bonds that correspond to the pair of 3-site objects $\{i, j, k\}$ and $\{l, m, n\}$. To make explicit what we mean by "appear", take the case of $\sigma_i^\mu \sigma_j^\nu \sigma_k^\gamma$. If the bonds $\{\langle i, j \rangle, \langle j, k \rangle\}$ host $\{\sigma_i^x \sigma_j^x, \sigma_j^z \sigma_k^z\}$, then $\sigma_i^\mu \sigma_j^\nu \sigma_k^\gamma$ must equal $\sigma_i^x \sigma_j^x \sigma_j^z \sigma_k^x = -i\sigma_i^x \sigma_j^y \sigma_k^x$ for any correlator involving this 3-site operator to be non-zero. If the bonds $\{\langle i, j \rangle, \langle j, k \rangle\}$ host $\{\sigma_i^x \sigma_j^x, \sigma_j^x \sigma_k^x\}$, then $\sigma_i^\mu \sigma_j^\nu \sigma_k^\gamma$ must equal $\sigma_i^x \sigma_j^x \sigma_j^x \sigma_k^x = \sigma_i^x \sigma_j^0 \sigma_k^x = \sigma_i^x \sigma_k^x$ which is actually a 2-spin operator for any correlator involving this "3-site" operator to be non-zero. And so on. A similar multiplication rule can be followed to arrive at higher-spin operators with non-zero correlations. The above structure is intuitive as well since it accords with the short-range correlations that the Hamiltonian terms are trying to favor in the system. E.g. for the 2-spin operators, it is precisely the bond energies which have non-zero correlations, etc.

All of the above proofs can be extended to dynamical correlators as well. In the dynamical case, e.g. for 2-point correlators which is sufficient to illustrate the main point, we have $\langle \sigma_i^\mu(t) \sigma_j^\nu(0) \rangle = \langle e^{-iHt} \sigma_i^\mu e^{iHt} \sigma_j^\nu \rangle$. We recall here that the ground state expectation value is again an equal mixture over the ground state manifold (Eq. 27). Since 1) the subdivision of the ground state manifold into disjoint sets is based on the conserved quantities of the Hamiltonian, and 2) the conserved quantities will commute across the $e^{-iHt}$ and $e^{iHt}$ factors by definition, the cancellation argument will also apply to the dynamical case as well. Table 1 will essentially be reproduced for finite $t$ after replacing "$\sigma_{i_x}^\mu \sigma_{i_x}^\nu$" with "$\sigma_{i_x}^\mu(t) \sigma_{i_x}^\nu(0)$" everywhere. Thus the sum over the ground state manifold in each disjoint set contributing towards $\sigma_{i_x}^\mu(t) \sigma_{i_x}^\nu(0)$ will again cancel out to zero. Similarly, all the arguments in the paper for static multi-spin correlators extend to corresponding dynamical multi-spin correlators as well.

Finally, we end this section by considering how the above arguments apply to the one-dimensional models of Sec. 1.1 with the representative example of Eq. 1. Even though there is an extensive degeneracy in this case, the conserved quantities $\sigma_i^x \sigma_{\partial i}^x$ can not make the ground state expectation $\langle \sigma_i^x \sigma_j^x \rangle$ vanish. All other ground state expectation $\langle \sigma_i^\mu \sigma_j^\nu \rangle$ with $\mu \neq x$ or $\nu \neq x$ do vanish by the above kind of arguments. We can however use the above kind of

Table 2: A generic set $A$ of 4 ground states that form an equivalence class given two sites $\partial i$ and $\partial j$ using the conserved $\{\sigma^z_{\partial k}\}$ basis for the model of Eq. 1 and Fig. 1b.

| $|\psi\rangle$ | $\langle\psi|\sigma^z_{\partial i}|\psi\rangle$ | $\langle\psi|\sigma^z_{\partial j}|\psi\rangle$ | $\langle\psi|\sigma^\mu_{\partial i}\sigma^\nu_{\partial j}|\psi\rangle$ |
|---|---|---|---|
| $|\psi^A_{\text{gs}}\rangle$ | +1 | +1 | $\langle\psi^A_{\text{gs}}|\sigma^\mu_{\partial i}\sigma^\nu_{\partial j}|\psi^A_{\text{gs}}\rangle$ |
| $\sigma^x_i\sigma^x_{\partial i}|\psi^A_{\text{gs}}\rangle$ | -1 | +1 | $\left(2\delta_{\mu x}-1\right)\langle\psi^A_{\text{gs}}|\sigma^\mu_{\partial i}\sigma^\nu_{\partial j}|\psi^A_{\text{gs}}\rangle$ |
| $\sigma^x_j\sigma^x_{\partial j}|\psi^A_{\text{gs}}\rangle$ | +1 | -1 | $\left(2\delta_{\nu x}-1\right)\langle\psi^A_{\text{gs}}|\sigma^\mu_{\partial i}\sigma^\nu_{\partial j}|\psi^A_{\text{gs}}\rangle$ |
| $\sigma^x_i\sigma^x_{\partial i}\sigma^x_j\sigma^x_{\partial j}|\psi^A_{\text{gs}}\rangle$ | -1 | -1 | $\left(2\delta_{\mu x}-1\right)\left(2\delta_{\nu x}-1\right)\langle\psi^A_{\text{gs}}|\sigma^\mu_{\partial i}\sigma^\nu_{\partial j}|\psi^A_{\text{gs}}\rangle$ |

Table 3: A generic set $A$ of 4 ground states that form an equivalence class given two sites $\partial i$ and $\partial j$ using the conserved $\{\sigma^x_{\partial k}\sigma^x_{\partial k}\}$ basis for the model of Eq. 1 and Fig. 1b.

| $|\psi\rangle$ | $\langle\psi|\sigma^x_i\sigma^x_{\partial i}|\psi\rangle$ | $\langle\psi|\sigma^x_j\sigma^x_{\partial j}|\psi\rangle$ | $\langle\psi|\sigma^\mu_{\partial i}\sigma^\nu_{\partial j}|\psi\rangle$ |
|---|---|---|---|
| $|\psi^A_{\text{gs}}\rangle$ | +1 | +1 | $\langle\psi^A_{\text{gs}}|\sigma^\mu_{\partial i}\sigma^\nu_{\partial j}|\psi^A_{\text{gs}}\rangle$ |
| $\sigma^z_{\partial i}|\psi^A_{\text{gs}}\rangle$ | -1 | +1 | $\left(2\delta_{\mu z}-1\right)\langle\psi^A_{\text{gs}}|\sigma^\mu_{\partial i}\sigma^\nu_{\partial j}|\psi^A_{\text{gs}}\rangle$ |
| $\sigma^z_{\partial j}|\psi^A_{\text{gs}}\rangle$ | +1 | -1 | $\left(2\delta_{\nu z}-1\right)\langle\psi^A_{\text{gs}}|\sigma^\mu_{\partial i}\sigma^\nu_{\partial j}|\psi^A_{\text{gs}}\rangle$ |
| $\sigma^z_{\partial i}\sigma^z_{\partial j}|\psi^A_{\text{gs}}\rangle$ | -1 | -1 | $\left(2\delta_{\mu z}-1\right)\left(2\delta_{\nu z}-1\right)\langle\psi^A_{\text{gs}}|\sigma^\mu_{\partial i}\sigma^\nu_{\partial j}|\psi^A_{\text{gs}}\rangle$ |

arguments to prove the (classical) spin liquidity on the auxiliary partner sites. Working in the conserved $\{\sigma^z_{\partial i}\}$-basis, we arrive at Table 2, while working in the conserved $\{\sigma^x_i\sigma^x_{\partial i}\}$-basis, we arrive at Table 3. Combing both of them, $\langle\sigma^\mu_{\partial i}\sigma^\nu_{\partial j}\rangle = 0$ for any $\mu$, $\nu$ as was already indicated in our interpretation in Sec. 1.1.

## 2.3    Comparison and contrast with known quantum spin liquids

Since the only non-zero mean-fields are those that correspond to the short-range correlations induced by the Hamiltonian, this suggests the absence of other kinds of non-magnetic spontaneous symmetry breaking as well. Long-range magnetic correlations are not present as we have seen earlier with 2-point correlation arguments for 1-spin operators. This is emblematic of a quantum spin liquid. The exact nature of the spin liquids represented by the models of Eq. 23 and Eq. 26 is not clear due to a lack of an exact solution. Certainly a Kitaev or Jordan-Wigner like free-fermionization is not operative here. We can also already say that they are different than Kitaev spin liquids due to the extensive entropy and the anticommutation structure.

It remains to be seen whether this spin liquid is gapped or gapless modulo the extensive zero modes, even though the above already implies that spin correlations are extremely short-ranged, since there can be fractionalized excitations in this model analogous to Kitaev honeycomb model. The absence of other symmetry breaking orders further suggests fractionalization. This is an open question. We conjecture that the spectrum of Eq. 23 and Eq. 26 is gapless when all couplings are equal ($J_x = J_z$) and gapped otherwise modulo the extensive zero modes. A solvable one-dimensional example in the form of the chain limit of the models discussed above is given in Sec. 3.3 to support this conjecture.

The degeneracies of these models violate the bound on degeneracy of homogeneous topological order (38) possibly signaling an interpretation of the ground state manifold in terms of fractionalized zero modes and, more generally, gapless fractionalized excitations (39). Could it be that these models entirely evade a quasiparticle description to accord with the extensive

ground state entropy similar to the fermionic SYK model? Here it may be remarked that in Kitaev honeycomb model which admits a Majorana quasiparticle description, 1-spin correlations (2-point correlators) have the hyperlocal property as a consequence of fractionalization of the spins into Majorana and $Z_2$-gauge degrees of freedom (36), long-distance 2-spin correlations (4-point correlators) are non-zero, decaying exponentially in the gapped phase and algebraically in the gapless phase. The hyperlocal nature of 2-point correlators is also true for models with the «anticommuting» structure as proved in the previous section (Sec. 2.2). The higher multi-spin correlators that survive the cancellations from the the «anticommuting» mechanism may have similar long distance properties as the Kitaev honecomb model.

There is another $O(1)$ "degeneracy" (40) that we did not touch upon in the previous Subsec 2.1 that is easier seen in the model of Eq. 23. There are non-local string operators which are conserved similar to TFQIM and Eq. 25 for this model (and also Eq. 26). For Eq. 23, they may be written as $\prod_{r_y} \sigma^x_{(r_x,r_y)}$, $\prod_{r_y} \sigma^y_{(r_x,r_y)}$, $\prod_{r_y} \sigma^z_{(r_x,r_y)}$ on a horizontal row of sites, and $\prod_{r_x} \sigma^x_{(r_x,r_y)}$, $\prod_{r_x} \sigma^y_{(r_x,r_y)}$, $\prod_{r_x} \sigma^z_{(r_x,r_y)}$ on a vertical row of sites. They can be moved from row to row by the multiplication of appropriate plaquette conserved quantities of this model without changing the conserved value of the string operators. Only two of them can be chosen to be mutually commuting and form four global parity "superselection" sectors. Such a structure is also present in Kitaev honeycomb model (41). We may take them to be $\left(\prod_{r_y} \sigma^\mu_{(r_x,r_y)}, \prod_{r_x} \sigma^\mu_{(r_x,r_y)}\right)$ for $\mu \in \{x,y,z\}$. Another choice could be $\left(\prod_{r_y} \sigma^x_{(r_x,r_y)}, \prod_{r'_y} \sigma^z_{(r_x,r'_y)}\right)$ where $r_y$ may or may not equal $r'_y$. Within each global parity superselection sector, there will be an extensive entropy due to the anticommuting mechanism. The local conserved quantities can not change the values of the conserved global parities. Even though this smells like a topological degeneracy (as the thermodynamic limit is approached similar to TFQIM), it is not clear if the ground state manifold should be considered topologically ordered in the sense of the Kitaev toric code (which obeys Haah's bound on homogeneous topological order (38)) due to the exponentially large ground states in each global superselection sector.

Another point of contrast with Kitaev toric code is that due to the mutually conserved nature of the terms in the Kitaev toric code, they can be used to work out the ("$e$" and "$m$") excitations (with mutual anyonic phase of $\pi$). This allows to explicitly see the non-local operations that can be done – creating a pair of excitations and annihilating them after taking one of them across a non-trivial loop on the torus – to change the topological sector (26). In our case, the nature of the non-local operations needed to change the superselection sector is not clear. Note that the using the local conserved quantities of Eq. 23 do not help in this since they also commute with the string operators and do not change their values. Rather they give rise to the extensive entropy in each superselection sector through the anticommuting mechanism.

For Eq. 25 and Fig. 4, we do not have an extensive degeneracy, but rather a double degeneracy. This is to be thought of as being similar to the double degeneracy of TFQIM on the Ising ordered side. Thus this model is rather Ising ordered with the order being in $x$-direction or $z$-direction depending on the relative magnitudes of $J_x$ and $J_z$ which is intuitive as well. Note there is a sub-extensive degeneracy at the transition point $J_x = J_z$ (31), which is indicative of spin liquidity at this point. Even though free fermionization is not operative for this model, one can make a mean-field argument for a Majorana liquid at the quantum phase transition. One can do a Jordan-Wigner transformation of Eq. 25 using a snake-like Jordan-Wigner string (17) to arrive at

$$H = J_x \sum_{\langle \mathbf{r}, \mathbf{r+e_x}\rangle} c^\dagger_\mathbf{r} c_\mathbf{r+e_x} + c^\dagger_\mathbf{r} c^\dagger_\mathbf{r+e_x} + \text{ h.c. } + J_z \sum_{\langle \mathbf{r}, \mathbf{r+e_z}\rangle} \left(n_\mathbf{r} - \frac{1}{2}\right)\left(n_\mathbf{r+e_z} - \frac{1}{2}\right) \tag{28}$$

Performing a mean-field decoupling of the four-fermion term and assuming zero Ising magne-

tization at the transition, one arrives at

$$H_{\mathrm{mf}} = J \sum_{\mathbf{r}} c_{\mathbf{r}}^\dagger c_{\mathbf{r}+\mathbf{e_x}} + c_{\mathbf{r}}^\dagger c_{\mathbf{r}+\mathbf{e_x}}^\dagger + c_{\mathbf{r}}^\dagger c_{\mathbf{r}+\mathbf{e_z}} + c_{\mathbf{r}}^\dagger c_{\mathbf{r}+\mathbf{e_z}}^\dagger + \text{ h.c.} \qquad (29)$$

where $J = J_x = J_z$. This is a $p$-wave superconductor of spinless fermions with gapless nodes in the two dimensional Brillouin zone with Majorana excitations, very analogous to the TFQIM transition in one dimension.

# 3 Conclusion

## 3.1 Summary of results

This work describes a construction for spin-$\frac{1}{2}$ models which results in an extensive residual ground state entropy and quantum spin liquidity. For any Hamiltonian, if it hosts an «anticommuting» algebra of local conserved quantities that have extensive cardinality, such behaviour would manifest. The discussion around Eq. 23 and Eq. 26 in Sec. 2 gives examples of this «anticommuting» structure or mechanism. The extensive entropy property and the concomitant spin liquidity holds throughout the full eigenspectrum. This construction is natural in the presence of bond-dependent couplings and the resultant physics is exposed through the examples of the models in Eq. 23/Fig 3 and Eq. 26/Fig 5.

The basic aspects of extensive entropy was exposed through pedagogical one-dimensional examples in Sec. 1.1. The main example of Eq. 1 as a variant of the well-known transverse field quantum Ising model had effectively classical conserved (auxiliary) spins which did not void quantum (Ising) order. In Sec. 2, we constructed models without any effective classical degrees of freedom. This led to provable quantum spin liquidity (Sec. 2.2) apart from the extensive residual entropy (Sec. 2.1). The hyperlocal vanishing of the static and dynamical spin 2-point correlators is a natural consequence of the local «anticommutation» structure and the resulting spectral degeneracies. This is in fact true throughout the spectrum and thus for all temperatures. A detailed comparison and contrast of these quantum spin liquids to other known quantum spin liquids and models was given in Sec. 2.3.

Another aspect is the gauge-like nature of the extensively many «anticommuting» local conserved quantities in these constuctions.. This may be a novel way in which gauge-like physical degrees of freedom emerge in quantum spin-$\frac{1}{2}$ systems, e.g. when comparing to the Levin-Wen model and Kitaev's toric code, Kitaev's honeycomb model, Haah's code and X-cube model (26; 42; 43; 44; 45) all of which have only commuting conserved sets. This anticommuting or non-commuting mechanism can in general operate in any number of dimensions. A natural construction is similar to the model of Eq. 23 on the three-dimensional pyrochlore lattice. A two-dimensional version of this on the "two-dimensional pyrochlore lattice" is shown in Fig. 6. Also may be mentioned that the anticommuting mechanism seems natural for spin models, in particular spin-$\frac{1}{2}$, since bosons do not naturally accommodate anticommutation and it does not appear so for fermions as well (46). Higher spin models can accommodate more general forms of non-commutation and local versions of non-commutation beyond anticommutation will be interesting to find.

## 3.2 Remarks on Physical consequences and realization

The physical consequences originating from the «anticommuting» mechanism in a particular two-dimensional model not discussed here and closely related to case (a) in Sec. 1.1 has been discussed in Ref. (17). In general, we expect a residual ground state entropy even when the excitation spectrum is gapped due to the degeneracy of the ground state manifold. This can

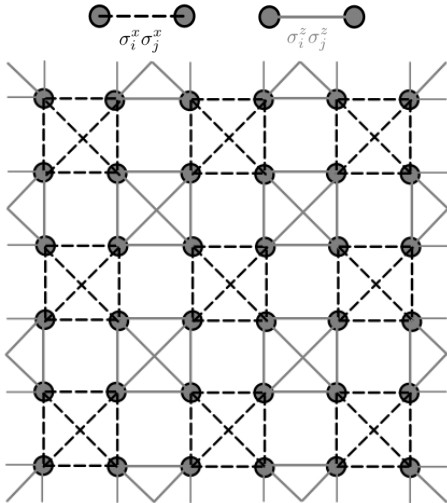

Figure 6: A model on the two-dimensional analog of the three-dimensional py-
rochlore lattice with extensive ground state entropy enforced by the anticommuting
mechanism. Analogous results from the previous Sec. 2 will apply to this model as
well.

lead to a two-peak structure in the magnetic component of the specific heat similar to Kitaev
spin liquids (17). The hyperlocal nature of 2-point spin correlators proved in Sec. 2.2 for
the higher-dimensional quantum constructions of Sec. 2 implies no magnetic ordering and
completely featureless spin structure factors at all energies. Spin structure factors of quan-
tum magnets are commonly probed in inelastic neutron scattering experiments. The presence
of global superselection sectors discussed in Sec. 2.3 leaves open the intriguing prospect of
topological order coexisting with quantum spin liquidity.

In terms of experimental realizations, artificial quantum systems seem to be the best bet
for observing the physics of these models. One such example is Ref. (47) which proposes a su-
perconducting qubit based analog simulation of bond-dependent lattice spin model physics. In
a solid state material context, perhaps some quantum compass type lattice realization (28; 29)
may accommodate the needed structure of the couplings as an alternative to the Jackeli-
Khaillulin mechanism (48) for Kitaev type bond-dependent couplings. Another possibility is
that the basic physics of these models can be made operational in the low energy physics of
some quantum spin-$\frac{1}{2}$ ice materials (10) if one arranges for the quantization axes of the quan-
tum Ising couplings to "stagger" across the lattice in some sensible way. Quantum fluctuations
in quantum spin ices are usually induced through a transverse field term as opposed to the
anticommuting mechanism. A Rydberg-atom based artificial implementation of quantum spin
ices has recently been proposed recently in Ref. (49) which can provide another direction for
an experimental realization of these models.

## 3.3 Open questions

1. A detail-oriented issue is if there are there additional "accidental" degeneracies at fine-
   tuned ratios of the couplings apart from the generic exponential degeneracies in the
   spirit of Ref. (31).

2. A general question is regarding the nature of excitations in these models. Is there a finite
   energy gap above the ground state manifold or not? Are there fractionalized excitations
   in models with the anticommutation structure and concomitant extensive entropies such

as the ones discussed in this paper? Could they entirely evade a quasiparticle description? tion?

One example where we can explicitly work out the nature of the excitations is a one-dimensional Kitaev chain model that can be considered a cousin of either the Kitaev honeycomb model or the models in Eqs. 23 and 24, i.e.

$$H = J_x \sum_{-x} \sum_{\langle i,j \rangle \in -_x} \sigma_i^x \sigma_j^x + J_z \sum_{-z} \sum_{\langle i,j \rangle \in -_z} \sigma_i^z \sigma_j^z \tag{30}$$

The above model has received attention in the literature (50; 51) in the context of Kitaev spin liquid physics. The «anticommuting » sets of local conserved quantities are

- $\sigma_i^z \sigma_j^z$ on $-_x$
- $\sigma_i^x \sigma_j^x$ on $-_z$

which again guarantees an extensive ground state entropy. The above Hamiltonian can also be reduced to a fermionic quadratic form using Majorana operators (52) to yield

$$H = J_x \sum_{-x} \sum_{\langle i,j \rangle \in -_x} u_{ij}^x \, \gamma_i \gamma_j + J_z \sum_{-z} \sum_{\langle i,j \rangle \in -_z} u_{ij}^z \, \gamma_i \gamma_j \tag{31}$$

where $\gamma^\dagger = \gamma$ and $u_{ij}^\mu$ are the $Z_2$-conserved quantities in the Kitaev representation ($\propto \gamma_i^\mu \gamma_j^\mu$), whose spectrum can be explicitly worked out (Eq. 32 of Ref. (27) with $J_y = 0$). It is nothing but an edge of the famous triangle phase diagram of the Kitaev honeycomb model (Fig. 5 of Ref. (27)). The excitation spectrum consists of Majorana modes that are gapped for $J_x \neq J_z$ and become gapless at the quantum phase transition $J_x = J_z$. Note the two phases on *both* sides have the same topological order describable by the toric code. Furthermore, each mode is necessarily extensively degenerate even when gapped. This is thus a violation of Haah's bound (38) for gapped topological order. Whether this is more than a pathology remains to be seen. However this is certainly natural when seen as a one-dimensional limit of Eq. 23. The extensive ground state entropy can also be interpreted as "physicalizing" ($Z_2$) gauge artifacts in an infinitely long one dimensional chain! (17).

In light of the above discussion, where we could pinpoint the nature of the excitations in the solvable one-dimensional case, one may wonder how can these models evade a quasiparticle description as conjectured before. This may still be the case in terms of the dynamical correlations. Even though the spectrum has a Majorana quadratic form in each block of Hamiltonian in the above example, time evolution will "instantaneously" start mixing the blocks, for *generic* initial states, which may effectively render a quasiparticle description ineffective for dynamical correlations.

3. In presence of deformations that void the anticommutation structure, what would be a generic consequence in models close to the limit where the anticommuting structure remains intact? An example of this was seen in Sec. 1.3 where the deformation was one of the set of conserved quantities. If the deformation is not one of the conserved quantities, does that generically imply the appearance of slow modes made out of the degenerate manifold as conjectured in Ref. (17), similar to what happens at the sub-extensively degenerate quantum phase transition in the 90° compass model (31)

4. The entanglement structure in the ground state manifold is certainly worth investigating. Can there be a way to make progress using the anticommutation structure without knowing the exact ground state solutions? Numerica will already have things to say about this issue.

5. Does there exist a statistical physics like perspective on the ground state manifold structure. Are there other physical interrelations between the ground states more than what the anticommutation structure stipulates, or at least a more detailed view of them? A classical example of this would be from constrained statistical physics models, e.g. the relation between the different classical spin ice ground states as being connected by loops where the spin orientations are flipped to connect them. Without the knowledge of the exact ground state structure, this is not obvious.

6. Of course, all of the above motivates constructing solvable cousins of these models. Constructions which are solvable and have extensive entropy can be written down, but it is not evident how to avoid solvable constructions which do not have any effective classical variables (conserved $\sigma_i^\mu$ for some $\mu$ and subset of sites). One such construction has been discussed in Ref. (17). Constructions that do not have any such effective classical degrees of freedom, host a generically extensive ground state entropy through the «anticommuting» mechanism as the models discussed in this work, and are solvable through some means would be very interesting to study and is an open question.

7. At a framework level, this work suggests a general theory for constructing models with extensive ground state entropy in the spirit of what Refs. (53; 54) and related papers (55; 56; 57; 58; 59; 60) do for spin models with free fermion spectra (61).

8. Finally it is not fully clear how does the strongly correlated physics described here fit in the atlas of strongly correlated physics such as many-body topological orders and/or the absence of quasiparticle descriptions.

## 3.4 Further Speculations

The gauge-like aspect of these models mentioned before in Sec. 3.1 deserves more exploration it seems to the author. For example, the interpretation of the extensive ground state entropy in the one dimensional Kitaev chain model as "physicalizing" ($Z_2$) gauge artifacts in an infinitely long one dimensional chain alluded to in the second point of Sec. 3.3; does this interpretation somehow extend to higher dimensions? A related quasi-one-dimensional example involving thin strips in the existing literature is Ref. (62) where the authors discuss the extensive entropy generation to be related to the recent developments under the rubric of "higher-form" symmetries (63). One difference is that non-commuting conserved operators are on system-width spanning long strings in the model of Ref. (62), whereas in the constructions discussed here, the non-commuting quantities are local throughout in this sense with support over $O(1)$ lattice sites. It remains to be seen if there is a higher-form symmetry perspective on the «anticommuting» mechanism that may inform further on this issue.

Another question is if there exists a field-theoretic formulation of these models in the continuum analogous to Chern-Simons field theories for many-body topological orders? A remark here would be that Landau levels also have an extensive degeneracy, however proportional to the system area and, unlike this paper, not exponentially large. Chern-Simons theories give a field-theoretic understanding of the resultant many-body insulating states with topological order (64). However time reversal is explicitly broken in these cases due to the presence of a magnetic field. In all the models discussed in this work, time reversal is preserved since the Hamiltonians are composed of 2-spin terms and $\sigma_i^\mu \to -\sigma_i^\mu$ under time reversal for spin-$\frac{1}{2}$ degrees of freedom. Furthermore, from an algebraic point of view, the generation of degeneracy in Landau levels is due to large symmetry or quantum number generators which do not appear in the Hamiltonian which is different from the anticommuting mechanism. For the SYK model which can have an extensive ground state entropy, there exist dynamical (mean-)field theory formulations in the continuum (11).

The final physical point that is perhaps of relevance relates to quantum chaos bounds (65). It has been shown that the SYK model saturates this bound (66; 67; 68). Given the extensive ground state entropy of the SYK model (69) and the relation of zero modes to the saturation of the chaos bound (68; 70), it is tempting to conjecture that the spin models discussed here may also approach – perhaps saturate – the quantum chaos bound like the original Sachdev-Ye model (71). These models and in particular the two-dimensional ones of Eq. 23/Fig. 5 and Eq. 26/Fig. 5 may then provide spin models with *local* interactions that approach the quantum chaos bound with appropriate butterfly velocities (72). In this context, the speculation of the absence of a quasiparticle description made earlier in Sec. 2.2 may also be pertinent. Another speculation would then be if these models with spin-$\frac{1}{2}$ microscopic degrees of freedom or qubits connect somehow to black hole physics in analogy with the connection between the SYK model and charged black holes (66; 73). Could these models have something to say about quantum gravity analogous to the SYK models?

# Acknowledgements

Discussions with Ajit Balram, Arkya Chatterjee, Pieter Claeys, Darshan Joshi, Saptarshi Mandal and Roderich Moessner are gratefully acknowledged.

**Funding information** Funding support from SERB-DST, India (superseded by ANRF-DST established through an Act of Parliament: ANRF Act, 2023) via Grant No. MTR/2022/000386 and partially by Grant No. CRG/2021/003024 is acknowledged.

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
