# Peer review of "A theorem on extensive ground state entropy, spin liquidity and some related models"

_SciPost Physics Core_

## Round 1 · Referee Report · Anonymous (Referee 1) · 2025-2-25

Report

Reviewer report on the revised manuscript: A theorem on extensive ground state entropy, spin liquidity and some related models by Sumiran Pujari

The revised manuscript has addressed several concerns raised in my previous report. The author has significantly improved the manuscript in multiple areas (e.g. including a clearer proof of the main theorem and restructuring of the abstract), whereas some other concerns remain either only partially addressed or completely unresolved (e.g. connection to real-world systems or the concluding section). No further revisions are thus needed for the adequately addressed points 3. and 4. related to the proofs of the main theorem, the point 5. related to the imbalance of the abstract, and the point 7. concerned with readability of general audience. However, further improvements are still needed for other points from my previous reviewer report.

1. Experimental Realization
This point was only partially addressed when the author acknowledges that the studied models are largely theoretical and that their direct realization in real materials is unclear. The revision now shortly mentions possible implementations in artificial quantum systems and draws a tentative connection to quantum spin ice materials. While this is a little step forward, the discussion remains somewhat speculative and lacks concrete suggestions for experimental validation. Although the author is not aware of any direct material realization of the considered model systems with rather peculiar bond-dependent interaction terms, the manuscript should at least discuss specific magnetic materials with the same magnetic structure (e.g. the chain system in Fig. 1(a) can be regarded as a comb or branched chain, the chain system in Fig. 1(d) as a delta or sawtooth chain).

2. Connection to Existing Literature
Although the author has incorporated several new references to relevant prior works, I am still lacking some further literature related to real-world experimental systems specifically mentioned in the previous point.

6. Nature of the Spin Liquids
The author admits that they do not have a complete characterization of the spin liquids present in their models. While they argue that the anticommutation structure is sufficient to guarantee spin liquidity, the nature of these spin liquids remains uncertain. The revised manuscript still lacks a detailed analysis of whether these are gapless or gapped spin liquids and how they relate to known quantum spin liquid classifications. Although a full characterization may not be feasible, at least a qualitative discussion on potential features (e.g., spin gap, long-range entanglement, topological properties, etc.) could be included. If the author conjectures that the spin liquids in 2D are fundamentally different from those in 1D, this should be supported with additional reasoning.

7. Lack of Numerical Simulations
The revised manuscript does not include any new numerical simulations to verify its predictions though they might be quite useful to advance overall understanding of the spin liquid phases. For instance, the spin gap could be numerically calculated to distinguish between gapless and gapful quantum spin liquids (see the previous point 6.). At minimum, the manuscript should explicitly discuss what types of numerical tests could be performed to validate the theoretical claims.

9. Too Broad Summary
The final section has been completely reorganized to separate the key results from more speculative discussions. However, it still includes a broad range of topics (quantum chaos, SYK models, quantum gravity, etc.) that are only marginally related to the main results. The concluding part is generally too long and it overshadows the paper’s primary contributions. The conclusion should be better put into much more concise and condense form, which should be mainly based on the current subsections 3.1. and 3.2. The subsections 3.3 and 3.4 related to open questions and further speculations should be either deleted or discussed only very briefly within no more than 2-3 sentences rather than 2 full pages.

The revised manuscript has been substantially improved in terms of clarity, rigor, and structure. The author has successfully addressed several major concerns, but some important points still remain unresolved, particularly regarding the experimental realization, characterization of the spin liquids, the numerical verification of the results, and the highly speculative character of the concluding part. Hence, I recommend a further round of major revisions before considering the paper for publication.

Recommendation

Ask for major revision

---

## Round 1 · Referee Report · Anonymous (Referee 2) · 2025-2-25

Report

I appreciate the author’s efforts in revising the manuscript and addressing some of the concerns raised in my initial report. The additional explanations and new references have improved the clarity of certain aspects of the work. However, while significant text has been added, the revisions remain superficial, and the manuscript still does not address the core issues that warranted a major revision. Many of the fundamental concerns from my first report remain unresolved. Below, I outline the main outstanding issues.
• The manuscript remains highly abstract, with insufficient discussion of potential physical realizations
In my initial report, I pointed out that while the models presented are mathematically well-defined, they lack a clear connection to physical systems. The author acknowledges this limitation, stating that they are not an expert in quantum chemistry. While I appreciate the addition of citations related to artificial quantum platforms and quantum spin ice materials, the manuscript itself still does not provide any real discussion of how the proposed models could be physically realized or what physical mechanisms would favor their emergence.
A theoretical paper does not necessarily need to propose an experimental realization, but it should at least discuss in more detail the types of interactions or conditions that might give rise to such models in real-world materials. Simply citing other works does not substitute for a clear, self-contained discussion within the manuscript.
• The conclusion remains unbalanced and overly speculative
One of my main concerns in the first review was that the "Summary and Outlook" section was more of a collection of open-ended questions rather than a synthesis of key results. The author has responded by restructuring this section into separate subsections and explicitly labeling speculative discussions under "Further Speculations." However, this change has only exacerbated the issue rather than resolving it.
The conclusion should primarily summarize the findings and their implications. While it is natural to suggest open questions, the current version still gives the impression that much of the paper is left unresolved or open-ended. In particular, the “Further Speculations” section feels unnecessary and detracts from the clarity of the manuscript.
Suggested Revision: The conclusion should focus on a concise summary of the key findings, their significance, and potential directions for future research. The speculative discussions should be significantly reduced or, ideally, removed altogether.
• Despite the additions, the revisions are mostly superficial
Although the author has added significant text in response to the first round of reviews, the revisions do not engage with the deeper issues that were raised. Instead of fundamentally addressing the concerns, the manuscript has mostly expanded in length without substantial improvement in clarity or depth.
Many of the additional explanations feel like an attempt to add volume rather than address the core criticisms. For example, while the author has added a discussion of physical realizations by citing external works, they have not actually incorporated a meaningful discussion of how their models relate to real-world materials. Similarly, while the conclusion has been restructured, it remains speculative and lacks a clear synthesis of the results.

Final Verdict: The manuscript still requires major revision
Despite the additional content, the manuscript remains too abstract, lacks sufficient physical motivation, and does not adequately clarify its novelty. I recommend another round of major revisions to properly address these fundamental issues before the paper can be considered for publication.

Recommendation

Ask for major revision

---

## Round 1 · Author Response

I thank the Referees for their valuable comments and criticisms which has substantially improved the manuscript. I hope they notice the various improvements in the revised manuscript thanks to their feedback. I have already responded individually to each of the Referee reports in a point-wise manner to all their directed questions and comments. Below is an itemized list of all the major revisions that are part of this resubmission.

---

## Round 1 · List of Changes

1) The abstract has been rewritten to better reflect the contents and the emphasis of the paper. Emphasis has been removed from the spectral equivalence of the model with auxiliary spins in Eq. (1) and the transverse field quantum Ising model which was somewhat besides the point. Now the emphasis of the abstract (and also the main text) is on the extensive ground state entropy and quantum spin liquid aspects of the higher-dimensional constructions. The proof of the aforementioned spectral equivalence has however been elaborated as suggested by the Referees.

2) The conclusion has been rewritten and reformatted to have a more coherent structure. Now it is subdivided into different subsections addressing different aspects. There are dedicated subsections to summarize the various results, their physical consequences and possibilities for experimental realization. All the more speculative discussion related to quantum chaos and quantum gravity from the previous version are compiled in the final subsection which is explicitly titled "Further Speculations".

3) An additional result on the dynamical spin correlations has been added.
See this  [post](https://scipost.org/submissions/2407.06236v6/#comment_id5080) as well.

4) All the additional and majorly revised text has been highlighted in blue for ease of reference.

5) Several references suggested by the Referees have been added including some related to physical realization. These are highlighted with an "Added" note  in red in the bibliography.

6) Regarding the suggestion of numerical simulations by the Referee in Report #1 to understand better the nature of these quantum spin liquids, apart from the responses made earlier, the discussion in point 2 of Sec. 3.3 on open questions addresses this issue directly by providing rather a spectrum-solvable analytical example where we can get a glimpse of the nature of these quantum spin liquids in one dimension in a fairly explicit way.

---

## Round 2 · Referee Report · Anonymous (Referee 1) · 2025-6-2

Report

The author has satisfactorily revised the manuscript according to the first two points of my second reviewer report. However, the points 3.-5. related to the nature of the spin liquids, the lack of numerical simulations, and too broad summary have not been addressed at all or only marginally.

The manuscript can be accepted for publication only after taking into consideration also those last three points or at least the rebuttal against them should be raised.

Recommendation

Ask for minor revision

---

## Round 2 · Referee Report · Anonymous (Referee 2) · 2025-6-2

Report

I appreciate the author’s continued revisions and acknowledge minor improvements, including brief comments on possible numerical investigations and the occurrence of certain lattice geometries in real materials. However, several concerns from my earlier reports remain only partially addressed.

The core issue of physical motivation is still unresolved. The manuscript lacks a clear discussion of what types of interactions or conditions might give rise to the proposed Hamiltonians in actual materials. The recent additions do not sufficiently engage with this point.

I also note that the author did not respond to the referee report. Even if there is disagreement, it would be appropriate to explain why certain suggestions were not implemented.

The speculative section remains too extensive. Although it has been renamed, the substance is largely unchanged. I continue to believe it should be significantly reduced or removed, as it distracts from the main results.

Recommendation:

I recommend minor revision to address the remaining issues and improve clarity.

Recommendation

Ask for minor revision

---

## Round 2 · Author Response

Dear Editor-in-charge,

Thank you for your assessment that the paper is suitable for Scipost Physics Core journal. I have done the minor revisions as requested and they are highlighted in blue for ease of reference. I believe the paper is ready for publication now and I hope you concur with this.

Best wishes,
Sumiran Pujari

---

## Round 2 · List of Changes

1) Discussion has been made to situate the one-dimensional lattice geometries in the context of existing material systems. 2) A separate point 5 has been mentioned now in Sec. 3.3 to indicate a list of numerical investigations that can be performed to understand these quantum spin liquids in more detail.

---

## Editorial Decision

awaiting_resubmission